# Attending a Blended In-Service Management Training in a Public Health System: Constraints and Opportunities for Pharmacists and Health Services

**DOI:** 10.3390/pharmacy9010012

**Published:** 2021-01-05

**Authors:** Andrigo Antonio Lorenzoni, Fernanda Manzini, Monica Cristina Nunes da Trindade, Bernd Heinrich Storb, Norberto Rech, Mareni Rocha Farias, Silvana Nair Leite

**Affiliations:** Department of Pharmaceutical Sciences, Universidade Federal de Santa Catarina, Florianópolis 88040-900, Brazil; andrigo.lorenzoni@posgrad.ufsc.br (A.A.L.); fernanda.manzini@posgrad.ufsc.br (F.M.); monica.nunes@posgrad.ufsc.br (M.C.N.d.T.); ppgasfar@contato.ufsc.br (B.H.S.); norberto.rech@ufsc.br (N.R.); mareni.farias@ufsc.br (M.R.F.)

**Keywords:** continuing education, e-learning, pharmacy workforce development, public health system

## Abstract

Management and public health are important domains of competency for pharmacists. In about 90% of Brazilian health departments, pharmacists manage the selection and purchase of medicines. The Pharmaceutical Services and Access to Medicines Management Course (PSAMM) was offered to pharmacists working in the public health system. The aim of this study is to analyze the impacts of the course as perceived by the students (pharmacists). Two thousand five hundred pharmacists completed the course. It is a mixed-methods study, including subscribing forms (*n* = 2500), evaluation questionnaire (*n* = 1500), focus groups (*n* = 10), and semi-structured interviews (*n* = 31). Participants reported a high level of satisfaction with the course; they considered to have developed competencies related to leadership and management, competencies needed to enhance and sustain their practices in health services. Data analyses showed important barriers to complete the course: high course workload, poor quality of Internet access, lack of support from the health services. Participants highlighted crucial features of the course that helped them develop key competencies: practical in-service activities, useful and realistic contents, tutoring. These features helped participants overcome some important constraints described by them. The educational model described in this study was perceived as having a long-term impact on their behaviors and management practices in health services.

## 1. Introduction

Access to medicines is an essential component to achieve universal health coverage [1]. However, access to medicines and their rational use are persistent global concerns. These issues have a significant impact on the health system’s quality and, consequently, on health [2]. The expansion of access to medicines was listed as one of the ten most prominent problems that demand attention from the World Health Organization [3]. It is necessary to address this issue from a health system perspective, considering the complex relationships between medicines and health financing, human resources, health information, and the broader issue of access to health services and interventions [1,4].

In Brazil, access to medicines is a responsibility defined by public policy. The National Pharmaceutical Policy defines services and responsibilities at all levels of governance of the unified health system (*Sistema Único de Saúde—SUS*) to guarantee access to medicines and pharmaceutical services.

With the decentralization of the Brazilian health system, the federal government is responsible for the general policies and guidance strategies. The 5560 municipalities, for their part, undertake a series of responsibilities that demand the mobilization of knowledge and technical, managerial, and political skills relative to medicines and services delivery [5]. Citizens have the constitutional right to access medicines, and about 47% of the population access essential medicines. Such access is ensured by a network of widely distributed public healthcare centers [6,7].

In about 90% of municipalities’ health departments, pharmacists manage the selection, planning, purchase, and distribution of medicines [8]. More than 30,000 pharmacists work in public healthcare and are employed by the SUS. This number grew by 75% from 2008 to 2013 [9]. However, there are still failures in the organization of pharmaceutical services and management. In Brazil, some studies have suggested that the currently available pharmaceutical workforce’s professional skills do not correspond with the expected profile for work in SUS, mainly related to management and leadership skills [10,11].

Health systems governance and management are currently critical concerns in many countries [12]. In fact, not only pharmacists but a significant number of healthcare managers do not hold formal management qualifications. In many healthcare professional degrees, there is little, or no management training integrated into the curriculum. This may be extremely problematic, given that clinical and managerial competencies are vastly different [13].

“Organization and management” competencies are one of the four fields of professional competencies that pharmacists need, according to FIP [14]. They are related to the management of teams, supply chains, services, budgets, and procurements. Communication, leadership, and interprofessional collaboration also belong to the set of competencies for pharmacists. Nevertheless, the literature mostly focuses on leadership training and competencies of physicians, nurses, and public health specialists [15].

The Ministry of Health (MoH) identified the need for training pharmacists, in charge of the pharmaceutical policy and development of services in the municipalities, in the management of health systems (focused on pharmaceutical-related issues). The MoH requested one public university to develop a national course to address the need for management training. The Pharmaceutical Services and Access to Medicines Management Course (PSAMM) was the most important continuing education (CE) initiative for pharmacists ever developed in Brazil. It was funded by the MoH as part of the activities of the Brazilian Unified Health System Open University (UNA-SUS, a consortium of universities created by the MoH to meet the training and education needs of the SUS). The course was offered to pharmacists working in the public health system, using e-learning as the main approach [16].

It is essential to broadly assess the CE initiatives and their impact on healthcare organizations [17,18], including several methods and points of view. An extensive analysis of the development and implementation of the PSAMM showed that it was affected by internal elements such as its didactic project and support infrastructure and also by external elements such as the sociopolitical scenario in the health system and support from the health services heads [16]. This study aimed to analyze the impacts of a management training course, perceived by pharmacists, in the context of the Brazilian health system.

## 2. Materials and Methods

### 2.1. Environment and Context of the Course

The PSAMM targeted pharmacists (government employees), and it was offered free of charge through a blended approach composed of mostly e-learning and four face-to-face meetings. Besides financing pharmacist training during the same period, the Brazilian Ministry of Health was supporting other actions, such as infrastructure and workforce expansion to develop pharmaceutical services and access to medicines in diverse municipalities [19].

The PSAMM course had a total number of 4244 students enrolled. By the end of the course, 2500 students completed the course between 2012 and 2016 (1445 pharmacists in class A and 1055 pharmacists in class B); 303 gave up, and 1441 students failed the course. It is important to highlight that the results presented here include only students that completed the course. These pharmacists work in 1068 of the 5570 Brazilian municipalities, located in all the 27 federal states.

The PSAMM curriculum was prepared by 50 experts from 20 institutions from all over the country. The course didactic material was produced in printed and online versions for the Moodle platform [20]. Asynchronous activities such as reading texts, cartoons, video documentaries, and forums for each topic were offered. Every student needed to develop a training activity in service called the operative plan, based on Carlos Matus’ principles of strategic situational planning [21]. The aim of the operative plan was to develop management skills in real and current health services where the pharmacists work. These operative plans were available for the health centers where they were developed, and, in many cases, they were put into practice as reported by Portela et al. [22].

The PSAMM total workload was 375 h, distributed over three main themes, and carried out over 15 months (64 weeks). The curriculum is described in Table 1 [16]. Students were divided into 31 regional centers located in 17 Brazilian states, where the evaluations and four face-to-face meetings were held. The tutors (also pharmacists) were experienced professionals from all regions of the country and linked to the regional course centers. The course had 132 peer tutors (25 students per group).

By the end of the course, each student was expected to acquire: knowledge about management and pharmaceutical policy, self-confidence in developing management functions, motivation to work, and improved skills and behaviors that increase the managerial capacity of pharmaceutical departments [23,24].

### 2.2. Study Design

Considering a variety of data sources related to the students’ perceptions of the PSAMM course’s features and possible impacts, the study design took into account well-documented and extensively used models for evaluating continuing education in many educational contexts. The Kirkpatrick model [25] proposes four distinct levels as a sequence of ways to evaluate an educational program: (1) reactions (satisfaction with the learning intervention), (2) learning (the extent to which professionals change or improve attitudes, knowledge, skills, and/or self-efficacy as a result of attending the educational interventions), (3) behavior (the extent to which professionals’ learning was translated into their post-learning behavior or their professional performance), (4) results (can be seen as patients’ health outcomes resulting from the influence of learning interventions on professionals’ behavior changes). In the present study, results are considered the changes in management practices in place in the health services in which the students work.

The evaluation of educational interventions to effectively change professional behavior can be seen within the behavior change theories. The theoretical domains framework can be mapped in three domains: capability (the individual’s psychological and physical capacity to engage in the activity concerned—it includes having the necessary basic knowledge and skills); motivation (all those brain processes that energize and direct behavior, not just goals and conscious decision-making); opportunity (all the factors that lie outside the individual that make the behavior possible or prompt it) [26]. This model forms part of the behavior change wheel proposed by Michie et al. [27].

This study was designed with the assumption of the Brazilian Policy for health professionals’ permanent education in the SUS [28], i.e., that students, educational intervention, and workplace are all parts of a continuing education strategy. To perform the evaluation, the Kirkpatrick model helped define categories related to the satisfaction with the educational intervention; the perceived impact on students (related to their gain in knowledge and changes in professional behaviors); and the perceived impacts on health services management. The theoretical domains framework brought up the category of the students’ characteristics, particularly those related to the domains of capability, motivation, and the evaluation of the course associated with the domain of opportunity.

Taking such assumptions into account, the study groups the data into three categories: (a) students’ sociodemographic, educational, and professional characteristics; (b) evaluation of the infrastructure, contents, and teaching methods; (c) perceived impacts on the students’ behaviors and the management practices in health services.

### 2.3. Data Collection and Analysis

A mixed methods research agenda was followed to facilitate the inclusion of multiple students’ perspectives about the possible impacts of the course for them and for the health services in which they worked, including four data sources and data collection methods at different times, as described in Table 2. The partial studies were conducted separately, and the integration was performed at the interpretation phase only.

#### 2.3.1. Subscription Form

The analysis of the students’ registration forms on the PSAMM Course used the following data: age group, gender, distribution of graduates by geographic region, Type of hiring and government level. The data were processed using Microsoft Excel^®^ 2010 software and analyzed using the dynamic table tool. This study considered data from 2500 graduates.

#### 2.3.2. Likert-Scale Questionnaire

A questionnaire with closed questions was prepared based on the model proposed by Ruggeri et al. [29] for e-learning programs for health professionals. The adaptation and usefulness to the didactic resources of the Moodle platform, the technical and personal facilities and difficulties perceived for the incorporation of online activities, and the perception of learning management concepts and their applicability in the work process were evaluated. For the analysis, a 4-point Likert-scale was employed (I totally agree/I agree/I do not agree, nor disagree/I disagree). One thousand five hundred students from class B responded to the questionnaire during the last face-to-face meeting in 2014. They received printed copies of the questionnaire and answered it anonymously. The questionnaire was previously validated by the tutors.

The analyses were performed using descriptive statistics in the STATA program—SPSS—Version 2014 (College Station, TX, USA) Association analyses were performed. These were expressed through frequencies and central tendency measurement tests by Brazil’s geographic regions, through exact Fisher’s test. Differences of *p* < 0.05 were considered statistically significant.

#### 2.3.3. Focus Groups and Semi-Structured Interviews

Qualitative data collection happened through focus groups and individual interviews. It happened between August and October of 2018 (on average, three years after the students had finished the course). This last qualitative data collection intended to identify whether the results found at the end of the course were sustained. It is also meant to determine the context in which the students perceive the course’s late results.

It was carried out in 10 states across the country. The criterion for the selection of the states for the data collection was the density of resident students. The interviews took place in the state capitals and, whenever possible, in 1 smaller municipality in each state.

Ten students were selected in each state for the focus groups, and three to be individually interviewed. The students were randomly selected from the list of students from each state. They were assigned to the interviews or to the focus groups according to their availability to participate in the given activity. The students were contacted by phone and email. Whenever necessary, new students were randomly selected and contacted until the desired sample was achieved.

Guiding questions for the focus groups were used to address topics like the students’ experience attending the continuing education course and working at SUS, and the impact of the course for them and for the health services. Ten focus groups were carried out in the capital cities and lasted for around 1 h and 30 min. Each focus group had an average attendance of 6 participants.

The interviews addressed topics such as the choice of attending the course, the course structure and activities, the identification of changes in professional practices and work processes due to the course. Thirty-one interviews were conducted in 14 municipalities.

All focus groups and interviews were recorded and transcribed.

To safeguard participants’ identities, they were identified in the individual interviews by the letter E—for “interview”, in Portuguese—followed by the acronym for their geographical region (EN—interview conducted in the North region, ENE—Northeast, ECO—Center-West, ES—South, ESE—Southeast). Focus groups were identified as GF—for “focus group”, in Portuguese—followed by a number to differentiate the ten groups, e.g., GF1.

Two researchers (AAL and FM) reviewed all notes from the field, transcribed audio-recordings, and read these materials repeatedly before generating an initial codebook. This process allowed the researchers to gain a holistic understanding of the data and increase familiarity with the content, both manifest and latent. An inductive approach was used to conduct content and thematic analyses: identification of all relevant concepts and examples expressed; indexation of highlighted excerpts by theme and correlation with others; rearrangement of the highlighted excerpts, considering their contexts and specificities, in a coherent and understandable flow; mapping of the concepts, scope, and nature of the studied phenomenon. A deductive approach was then applied to identify manifest content related to the impacts on students’ behaviors and changes in service management practices, connecting the themes to a theoretical framework [30,31].

NVIVO 10 (Mairena del Aljarafe, Sevilla, Spain) was used to manage the data during the coding process and help organize themes. The highlighted excerpts per category were indexed and, considering their contexts and specificities, selected to exemplify each category.

Finally, the results of each partial study, including quantitative and qualitative finds, were analyzed together to identify possible convergent and divergent results and complementary finds. The discussion and conclusions consider the whole set of results.

### 2.4. Ethical Considerations

This study obtained permission from the Ethics Committee on Research with Human Beings (CAAE: 46912815.0.0000.0121). All participants were asked to provide written consent. The information was anonymized, and no personal identifiers were used during data analysis or publication.

## 3. Results

### 3.1. Students’ Sociodemographic, Educational, and Professional Characteristics

The profile of students who completed the course is shown in Table 3.

Most students are women (77%) and are between 25 and 39 years old (68%). More than 60% had already completed some postgraduate studies before taking the course, predominantly specialization courses, and 55.4% reported having no previous experience with the use of distance education in continuing education. A total of 75.2% point out flaws in their undergraduate courses for not focusing on SUS.

The Southeast region has the highest concentration of students who completed the course (36.7%), followed by the Northeast region (26.3%). Among the students, working at the municipal level (76.3%) and being hired through public examinations (73.5%) predominate. Most students had one job (51.1%), but 10.4% had three or more job contracts. This is related to the fact that 38.6% work more than 40 h a week.

The majority considered the workplace structure to be inadequate (64%) and considered the quality of Internet access at work to be good (51.6%). However, 35.3% consider access to be precarious or state they do not have access to the Internet.

The variables age group, gender, distribution of students by region, type and government level of employment refer to the 2500 students who completed the course. The educational level variables, perceived level of previous training focused on SUS, the number of work contracts, weekly workload, quality of Internet access in the workplace, the structure of the workplace, and previous use of distance education refers to 1500 students in class B.

### 3.2. Evaluation of the Infrastructure, Contents, and Teaching Methods

In total, 966 questionnaires of 1500 administered were considered valid, 44 (4.5%) in the North Region, 364 (37.8%) in the Northeast Region, 122 (12.6%) in the Midwest Region, 309 (32.0%) in the Southeast Region, and 127 (13.1%) in the South Region. The results are presented in Table 4.

Students from all regions rated the course very positively. The operative plan (an in-1500 service hands-on activity that was mandatory for all the students) and the tutoring in small groups of students (the same tutor following the group throughout the course) stood out as the most useful and important resources for learning. In all the country regions, the rate was high, but for the operative plan and the tutoring, there were significant differences in rates between regions.

The results of the assessment of infrastructure, contents, and teaching methods from the perspective of PSAMM Course students in focus groups and interviews are presented in Table 5. There are 2215 min of audio recordings.

The themes identified among the categories of analysis highlight important motivators to attend the course, as well as constraints that made it difficult to complete it, including characteristics of the course, but also of the workplaces and the environment. The themes that emerged in the category “configuration of the course to overcome the barriers encountered” describe the course’s characteristics that contributed in a crucial way so that students could complete the course.

### 3.3. Perceived Impacts of the PSAMM Course on the Students’ Behaviors and on Management Practices in Health Services

The closed questionnaire results are presented in Table 6, with 966 responses considered valid (the same as item 3.2).

Students from all regions assessed that the course provided behavioral changes for them and positive changes in management practices in their workplaces. There was a significant difference between the regions and an average disagreement of 5.4% about the conditions to develop leadership in their workplace.

The results of the evaluation of the impacts of the course from the perspectives of PSAMM Course’s students obtained in the focus groups and in the interviews are presented in Table 7.

## 4. Discussion

The evaluation of the course from the perspective of the students brings insights on strategic issues for the offer of continuing education (CE) for health professionals who are responsible for the practical implementation of an important public policy in the country.

Although the evaluation of the course was generally positive, students described some difficulties in attending it. The profile of students presents the same characteristics of the professional category in the country [32,33] and helps to understand the constraints found. The perceived precarity in public health contents during the undergraduate course and the rates of students who have already taken some kind of specialization or post-graduation courses to demonstrate some capacity, as well as motivation, to seek training and professional improvement. Regarding the opportunity to change professional behavior, considering the CE as a key element of this process, Table 3 and Table 5 bring essential elements that indicate substantial limitations for students related to workload and workplaces with no or precarious Internet access and support for CE.

Even though e-learning has provided a very important opportunity for these professionals to be able to access CE while staying in their own cities, access to good quality Internet (and sometimes also electricity) is still a challenge in remote regions [34]; as described, for instance, by students from the Amazon region. It is worth highlighting that CE attendance barriers for pharmacists worldwide are lack of time [35], cost, workplace, personal factors, and significant constraints related to geographical access [36].

In addition to the country’s structural difficulties, students also realized that there is a low incentive for education on the part of health institutions for pharmacists, which constrains opportunities for behavioral changes. This finding is corroborated by data found in the literature, which suggests that training for skills development, such as leadership, focuses mainly on doctors, nurses, and public health specialists [15].

In this sense, the pharmacists’ path in the search for better training goes through obstacles. As shown in the students’ profile, several pharmacists face a high workload, with two or three jobs as a barrier to the search for professional development. Data available in the literature [15] suggest that health professionals do not have protected time for professional training, which requires the use of resources such as distance learning. As a result, students take this responsibility individually; professional development invades their personal lives, which results in work overload.

The evaluation of the infrastructure, contents, and teaching methods of the PSAMM Course indicates that some characteristics of the course provided opportunities to overcome the constraints found in attending the course and also to sustain the results even three years later (Table 4, Table 5 and Table 7). They highlighted the importance of close tutoring with peer feedback, face-to-face meetings, and hands-on in-service activities to promote learning opportunities. Among the results in Table 5, it is possible to identify the importance of strategies to bring the course closer to the reality experienced by students and their needs, such as regional centers, contents related to the particularities of each region, and flexibilities related to accessing the platform and completing the proposed activities.

Forums were the teaching methods with the lowest evaluation rates (Table 4) and were not mentioned by students in the qualitative research years later. Some authors suggest that the distance modality may be more suitable for some students than for others, pointing to the blended method (online and face-to-face) as a better choice [29,37,38]. The results reported here show that the use of e-learning proved to be adequate, considering the particularities already mentioned. However, they point out the importance of face-to-face meetings during the course, suggesting that the blended model, with decentralized centers and regional tutoring, provides better conditions for learning. Even in the North (Amazon) and Northeast regions, where the infrastructure of health services and Internet access are of lower quality, students positively evaluated the characteristics of the course. Such positive characteristics are revealed in their impact on students and services.

Students perceived the impacts of the course at all levels defined by Kirkpatrick: from the satisfaction with the educational intervention to the impacts on their levels of knowledge, going through behavioral changes and the reflexes in the management practices of the services where they work (Table 6 and Table 7). It is crucial to highlight that the students noticed positive results when they were still in the last stage of the course and that these results persisted and became clearer after about three years of the course’s completion. This result allows us to conclude that the PSAMM course brought a good and sustained return for the investment made by the Ministry of Health.

In this study, it is possible to find evidence of consolidation of competencies, such as leadership and communication, for example, indicated by the FIP as necessary competencies for the pharmaceutical professional. These are fundamental to overcome the barriers found in the health system and to strengthen the role of the pharmacist in public health. The perception that the course had brought students better conditions to exercise leadership and conduct management tasks at the end of the course was, years later, consolidated into the concrete performance of these important competencies for a health services manager.

The changes reported by students in their behavior are in line with their reports about changes in service management practices, particularly in the use of information-based negotiation, with greater confidence and a broader view of the health system and how to overcome the barriers of hierarchy. The reports reveal the adoption of desirable and important management practices for the sustainability of health services, such as interprofessional collaboration and close relationship with popular local health councils, as well as strategic planning and agreement on actions related to access and rational drug use. These changes result in greater visibility and authority for the pharmaceutical sector within the healthcare system and for the pharmacist.

These impacts, however, happened and were reported at different intensities among the participants in this study. Contextual and policy issues related to local health services constrained the full development of behaviors and practices, as evidenced by the reported difference between the regions of the country in the ability to exercise leadership in their workplace (Table 7). The development of the operative plan was also constrained in some of the workplaces, particularly in regions North, Northeast and Midwest. Consequently, some students found the operative plan was not useful for them (Table 4 and Table 6). Hence, the intensity with which students report the achievements after the course must be weighted considering the context in which they attended the course and work.

As its main limitation, this study has an evaluation based on the perception reported only by pharmacists who completed the course, not including other points of view, such as managers and patients. Even so, the evidence presented here indicates that the provision of CE for pharmacists working in the public health system produces important and positive impacts for them and for health services.

The study also reveals issues that need to be considered when offering CE for health professionals. Students, educational intervention, and the workplace are drivers of the impacts of the CE. The Brazilian Policy for health professionals’ permanent education in the SUS [28] defines that educational interventions for health professionals must be understood and developed as a health system’s investment: therefore, educational activities must be part of health workers’ work, with the aim of achieving the health system’s benefits. Considering this assumption, the workplace support to develop educational intervention is an essential constituent of the CE strategy.

CE proposals must be attentive to the context of the target audience, proposing flexible strategies that make sense with the students and the local health system’s reality. New investments in continuing education should focus primarily on hands-on educational strategies with activities in the workplace, with compatible theoretical load, using the available resources [15], and meeting the needs of students; in a way that enables the development of the course at the service location and applying guidelines such as those reported by Ramani et al. [39] for the planning, implementation, and evaluation of the educational interventions in the local context.

## 5. Final Remarks

The set of results demonstrate the intrinsic relationship between the characteristics of students, the characteristics of educational intervention, and the impacts perceived by professionals/students in the practical reality of health services. Based on students’ perception, future educational proposals should rest primarily on strategies with a practical focus related to local services and pharmacists’ needs. Educational interventions’ design and health management must support pharmacists in overcoming the various constraints to attend CE.

The educational model described in this study had a long-term impact on students and services. As part of the Brazilian Ministry of Health investments to develop pharmaceutical services and access to medicines in diverse municipalities, continuing pharmaceutical education promotes important benefits in terms of pharmacists’ professional behaviors and management practices in health services.

## Figures and Tables

**Table 1 pharmacy-09-00012-t001:** Pharmaceutical services and access to medicines management course (PSAMM) course curriculum.

Transversal Module	Subject Matter	Modules	Time to Complete
Pharmaceutical services and access to medicine course	Health policies and access to medicines	Course introduction	75 h/10 weeks
	Access to medicines management and pharmaceutical services	Health policies and access to medicines	165 h/22 weeks
		Drug selection	
		Dispensing medicines	
	Operative plan development	Operative plan	60 h/during the course
	Complementary studies	Research methodology	75 h/10 weeks
		Special topics in ethics, evaluation of health technologies, technical and legal aspects related to allopathic medicines	
		Special topics in ethics, evaluation of health technologies, technical and legal aspects related to homeopathic medicines	
		Special topics in ethics, health education, therapeutic follow-up models	
Course completion work (about operative plan development)			12 weeks

**Table 2 pharmacy-09-00012-t002:** Categories and data sources.

Categories	Data Source
Students’ sociodemographic, educational, and professional characteristics	(1) Subscription form PSAMM course
	(2) Likert-scale questionnaire
Evaluation of the infrastructure, contents, and teaching methods	(1) Likert-scale questionnaire
	(2) 10 Focus group
	(3) 31 semi-structured interviews
Perceived impacts of the PSAMM course in the students’ behaviors and in the management practices in health services.	(1) Likert-scale questionnaire
	(2) 10 Focus group
	(3) 31 semi-structured interviews

Source: prepared by the authors.

**Table 3 pharmacy-09-00012-t003:** Students’ sociodemographic, educational, and professional characteristics—PSAMM course.

Variable	Items	%
Age range (*n* = 2500)	Up to 24 years	0.4
	Between 25 and 39 years	68.0
	Between 40 and 59 years	30.0
	60 and above	1.6
Gender (*n* = 2500)	Female	77.0
	Male	23.0
Distribution of graduates by geographic region (*n* = 2500)	Southeast	36.7
	Northeast	26.3
	South	19.2
	North	9.6
	Center-west	8.2
Labor relationship (*n* = 2500)	Government employee (with job stability)	73.5
	Government employee (without job stability)	26.5
Governmental level (work) (*n* = 2500)	Municipal	76.3
	State	18.7
	Federal	5.0
Educational level (*n* = 1500)	Undergraduate	34.0
	Specialization	55.6
	Master’s degree	8.6
	Doctorate degree	1.8
Training focused on SUS (*n* = 1500)	Bad/very bad	75.2
	Good/very good	24.8
Number of employment contracts (*n* = 1500)	One job	51.1
	Two jobs	38.5
	Three jobs or more	10.4
Weekly workload (*n* = 1500)	Up to 40 h per week	61.4
	More than 40 h per week	38.6
Internet access and quality in the workplace (*n* = 1500)	Very good	13.1
	Good	51.6
	Precarious	25.7
	No access	9.6
Workplace structure (*n* = 1500)	Not appropriate	64.0
	Appropriate	36.0
Previous use of distance education (*n* = 1500)	No	55.4
	Yes	44.6

**Table 4 pharmacy-09-00012-t004:** Answers to the questionnaire on the evaluation of the infrastructure, contents, and teaching methods—PSAMM course.

Issues Assessed	Likert Scale	Answers (Total (%))						*p*
		N	NE	CW	S	SE	Total	
Did the resources available in the virtual environment (Moodle) facilitate learning?	I totally agree	18 (40.9)	151 (41.5)	37 (30.3)	45 (35.4)	126 (40.8)	377 (39.0)	0.276 ^#^
	I agree	24 (54.5)	191 (52.5)	71 (58.2)	69 (54.3)	157 (50.8)	512 (53.0)	
	I do not agree nor disagree	0 (0.0)	6 (1.6)	2 (1.6)	3 (2.4)	4 (1.3)	15 (1.6)	
	I disagree	2 (4.5)	16 (4.4)	12 (9.8)	10 (7.9)	22 (7.1)	62 (6.4)	
Did the availability of the tutor contribute to the development of the activities?	I totally agree	29 (65.9)	264 (72.5)	78 (63.9)	94 (74.0)	230 (74.4)	695 (71.9)	0.092 ^#^
	I agree	11 (25.0)	91 (25.0)	39 (32.0)	29 (22.8)	75 (24.3)	245 (25.4)	
	I do not agree nor disagree	3 (6.8)	2 (0.5)	0 (0.0)	1 (0.8)	1 (0.3)	7 (0.7)	
	I disagree	1 (2.3)	7 (1.9)	5 (4.1)	3 (2.4)	3 (1.0)	19 (2.0)	
Was the “forum” tool effective for your learning process?	I totally agree	18 (40.9)	109 (29.9)	40 (32.8)	45 (35.4)	123 (39.8)	335 (34.7)	0.067 ^#^
	I agree	21 (47.7)	196 (53.8)	54 (44.3)	58 (45.7)	146 (47.2)	49.2	
	I do not agree nor disagree	0 (0.0)	12 (3.3)	10 (8.2)	7 (5.5)	13 (4.2)	42 (4.3)	
	I disagree	5 (11.4)	47 (12.9)	18 (14.7)	17 (13.4)	27 (8.7)	114 (11.8)	
Did the development of the operative plan help to learn the concepts related to the management of pharmaceutical services?	I totally agree	30 (68.2)	197 (54.1)	62 (50.8)	75 (59.1)	197 (63.7)	561 (58.1)	0.019
	I agree	12 (27.3)	160 (44.0)	57 (46.7)	52 (40.9)	110 (35.6)	391 (40.5)	
	I do not agree nor disagree	0 (0.0)	4 (1.1)	1 (0.8)	0 (0.0)	0 (0.0)	5 (0.5)	
	I disagree	2 (4.5)	3 (0.8)	2 (1.6)	0 (0.0)	2 (0.6)	9 (0.9)	
Do face-to-face meetings contribute to management learning?	I totally agree	22 (50.0)	191 (52.5)	50 (41.0)	64 (50.4)	143 (46.3)	470 (48.6)	0.322 ^#^
	I agree	17 (38.6)	144 (39.6)	56 (45.9)	48 (37.8)	139 (45.0)	404 (41.8)	
	I do not agree nor disagree	1 (2.3)	9 (2.5)	7 (5.7)	2 (1.6)	8 (2.6)	27 (2.8)	
	I disagree	4 (9.1)	20 (5.5)	9 (7.4)	13 (10.2)	19 (6.1)	65 (6.7)	

Sources: Brazilian Geographical regions: N, North; NE, Northeast; CW, Center-west; SE, Southeast; S, South. # *p*-value (exact Fisher’s test) after merging the two lines “I do not agree nor disagree” and “I disagree”.

**Table 5 pharmacy-09-00012-t005:** Analysis of focus groups and interviews on the evaluation of the infrastructure, contents, and teaching methods from the students’ perspective—PSAMM Course.

Analytical Categories	Themes	Selected Quotes
Opportunity to do EC	Course theme	“*A management education focused on pharmaceutical management, focused on the reality of SUS, is a theme that was not discussed in any form in my pharmacy undergraduate course*” [GF09]
		“*I had no experience at SUS, so he came a lot at the right time, the need to know more about how it works*” [GF09]
	Offer of the CE in the e-learning modality	“*It was impossible for me to be present, to have a physical presence*.” [ENE9]
		“*But currently, you either do e-learning or you don’t have the opportunity to do CE*.” [GF2]
	Free of charge	“*For the first time in my life, I would have a qualification without having to spend money, in the field in which I worked*.” [ENE9]
Barriers encountered by the students to complete the course	Internet	*“I was supervising in indigenous villages. The internet was a big nuisance*.” [GF01]
		“*Here the internet in the community was horrible. In my work I didn’t have internet, I didn’t have access. For us to have access to the course, we had to ask the manager*“ [EN1]
	PSAM Course Workloads	“*I was desperate when I got home and I had a lot of articles to read, there were a lot of things, I thought, “I won’t be able to handle it*.” [EN4]
		“*The course had many activities. I remember that I used the whole weekend to do the activities.*” [ES5]
	Student work overload	“*In the period when I was taking the Course, I worked in two jobs. My routine was a little heavy, I worked from midnight to 7:00 am in a pharmacy and worked from noon to 6:00 pm here. It was pulled.* “ [ESE4]
	Lack of work support	“*I was allowed to participate in the face-to-face meeting, but I had a discount on the wages of the double hours. You have a policy that encourages you, you have the Ministry of Health that offer, but they have internal legislation that makes it difficult*” [GF08]
Course configuration to overcome the barriers encountered	Course organization adapted to the regional needs	“*And I thought the most interesting thing was that you had this act of care with the students [providing the materials in a DVD/flash drive], understanding the situation we live in with the difficulty to access the internet*.” [GF01]
	Decentralized coordination in regional centers	“*We had very good support here in our town. The tutors and regional coordinators would help us when we had difficulties* “ [EN5]
	Tutoring	“*So, the tutoring part was very important, it directed us a lot and consolidated the thoughts and actions*.” [ENE7]
	Face-to-face meetings	“*The face-to-face meetings were important to clear up doubts, ask questions. There were teachers from the region and from outside. There were those exchanges.*” [GF8]
	Quality of teaching material	“*I liked the teaching material very much. It is a material that I use to this day, to study, to teach classes. I’ve used it many times. That’s a point that I consider extremely positive in the course*.” [GF09]
		“*I had everything I needed available on the platform, so it met my needs*.” [ES4]
	In-service activities—operative plan	“*It pushed us to build a way, make a diagnosis, discuss with all stakeholders, design proposals, revise proposals, listen. Apply the management tools.*” [ENE6]
		“*If it were just theoretical content, it would not achieve the objective as it did, at least for me. Because theoretical content you see in any postgraduate course*.” [ENE10]
	Flexibility of activities	“*My concern when I started was in relation to texts, I imagined that because it is e-learning, classes could be monotonous, with only texts to read. I have already taken courses like this, and I was very positively surprised by the layout of* (Moodle), *it made me want to continue studying*.” [PL6]

**Table 6 pharmacy-09-00012-t006:** Responses to the questionnaire on perceived impacts of the PSAMM course on the students’ behaviors and on management practices in health services.

Issues Assessed	Likert Scale	Answers (Total (%))						*p*
		N	NE	CW	S	SE	Total	
Did the course give you qualifications for service and policy management?	I totally agree	25 (56.8)	196 (53.8)	56 (45.9)	63 (49.6)	165 (53.4)	505 (52.3)	0.906
	I agree	19 (43.2)	162 (44.5)	65 (53.3)	63 (49.6)	140 (45.3)	449 (46.5)	
	I do not agree nor disagree	0 (0.0)	1 (0.3)	0 (0.0)	0 (0.0)	0 (0.0)	1 (0.1)	
	I disagree	0 (0.0)	5 (1.4)	1 (0.8)	1 (0.8)	4 (1.3)	11 (1.1)	
Did the course’s learning provide better conditions for exercising leadership in your workplace?	I totally agree	20 (45.5)	159 (43.7)	51 (41.8)	54 (42.5)	134 (43.4)	418 (43.3)	0.981 ^#^
	I agree	21 (47.7)	186 (51.1)	61 (50.0)	66 (52.0)	155 (50.2)	489 (50.6)	
	I do not agree nor disagree	0 (0.0	1 (0.3)	5 (4.1)	0 (0.0)	1 (0.3)	7 (0.7)	
	I disagree	3 (6.8)	18 (4.9)	5 (4.1)	7 (5.5)	19 (6.2)	52 (5.4)	
Do you consider yourself able to apply the knowledge and skills developed in the course to your reality?	I totally agree	25 (56.8)	163 (44.8)	46 (37.7)	50 (39.4)	138 (44.7)	422 (43.7)	0.239
	I agree	17 (38.6)	193 (53.0)	75 (61.5)	74 (58.3)	162 (52.4)	521 (53.9)	
	I do not agree nor disagree	0 (0.0)	2 (0.5)	0 (0.0)	0 (0.0)	0 (0.0)	2 (0.2)	
	I disagree	2 (4.5)	6 (1.6)	1 (0.8)	3 (2.4)	9 (2.9)	21 (2.2)	
Did the operative plan provide the application of the concepts learned in the work practice?	I totally agree	28 (63.6)	171 (47.0)	48 (39.3)	65 (51.2)	172 (55.7)	484 (50.1)	0.035 ^#^
	I agree	15 (34.1)	176 (48.3)	69 (56.6)	60 (47.3)	127 (41.1)	447 (46.3)	
	I do not agree nor disagree	0 (0.0)	5 (1.4)	1 (0.8)	0 (0.0)	0 (0.0)	6 (0.6)	
	I disagree	1 (2.3)	12 (3.3)	4 (3.3)	2 (1.6)	10 (3.2)	29 (3.0)	
Did the course’s learning offer you the tools so that you can mobilize different actors in your work for the management process?	I totally agree	22 (50.0)	169 (46.4)	45 (36.9)	65 (51.2)	156 (50.5)	457 (47.3)	0.104 ^#^
	I agree	21 (47.7)	182 (50.0)	66 (54.1)	55 (43.3)	135 (43.7)	459 (47.5)	
	I do not agree nor disagree	0 (0.0)	4 (1.1)	2 (1.6)	1 (0.8)	1 (0.3)	8 (0.8)	
	I disagree	1 (2.3)	9 (2.5)	9 (7.4)	6 (4.7)	17 (5.5)	42 (4.3)	
Did the course content make it possible to systematize and interpret information in the identification and prioritization of problems in your workplace?	I totally agree	24 (54.5)	191 (52.4)	55 (45.1)	63 (49.6)	158 (51.1)	491 (50.8)	0.205
	I agree	19 (43.2)	169 (46.4)	64 (52.5)	60 (47.2)	150 (48.5)	462 (47.8)	
	I do not agree nor disagree	0 (0.0)	0 (0.0)	1 (0.8)	2 (1.6)	0 (0.0)	3 (0.3)	
	I disagree	1 (2.3)	4 (1.10)	2 (1.6)	2 (1.6)	1 (0.3)	10 (1.0)	
Was the pedagogical content produced throughout the course adequate for the qualification of the pharmacist for management?	I totally agree	25 (56.8)	176 (48.4)	51 (41.8)	56 (44.1)	164 (53.1)	472 (48.9)	0.299 ^#^
	I agree	18 (40.9)	173 (47.5)	63 (51.6)	62 (48.8)	134 (43.4)	450 (46.6)	
	I do not agree nor disagree	0 (0.0)	4 (1.1)	5 (4.1)	3 (2.4)	2 (0.6)	14 (1.4)	
	I disagree	1 (2.3)	11 (3.0)	3 (2.5)	6 (4.7)	9 (2.9)	30 (3.1)	

Source: Brazilian Geographical Regions: N, North; NE, Northeast; CW, Center-west; SE, Southeast; S, South. ^#^
*p*-value (exact Fisher’s test) after merging the two lines “I do not agree nor disagree” and “I disagree”.

**Table 7 pharmacy-09-00012-t007:** Analysis of focus groups and interviews on perceived impacts of the PSAMM course on the students’ behaviors and on management practices in health services.

Perceived Impacts	Theme	Quotes
Changes in student attitudes/practices due to the course	Increase in Confidence	“*I am confident, I have more autonomy. The knowledge acquired was very important to consolidate the actions within the pharmacy, for nursing and for the direction of the service. I have more knowledge and the actions I propose here are more strengthened*” [ENE6]
	Communication capacity	“*We begin to see other possibilities, to listen to others. In management, it is too easy to make a decision disconnected from reality, but when we go to the service and look at the factual reality, we notice the need to put ourselves, at least a little, in other people’s shoes.* “ [ENE8]
	Negotiation capacity	“*This was for me the greatest merit of the course, it was teaching me what to do with the information that I have and that we can convince and we can show to* [the management].” [GF01]
	Leadership capacity	“*I went from being the pharmacy girl to effectively being the pharmacy service manager.*” [GF08]
	Change in the understanding of the role of the manager and of management	“*We stopped putting out fires on a daily bases to effectively become a work of management, of planning, in which you, according to the scenario in front of you, starts to program, plan, and look ahead in that scenario.*” [ESE5]
Benefits for health services	Negotiation with hierarchical superiors	“*To better negotiate with the suppliers, I needed access to the City Hall’s financial system. So, I negotiate the permission to access with the health secretary and the management secretary, and got good results.*” [ECO3]
	Interprofessional collaboration	“*We are pharmacists, and we work with nutrition, and we had never consider looking for a nutritionist to write the proper description of a product in an auction notice. So, we established a partnership with the nutritionists of the companies that already took part in the process to help us with the reference terms.*” [ECO3]
	Actions integrated with the popular councils of health	“*Participating in public audiences, participating in the Municipal Health Plan, participating in committees, all of this came after the course.*” [ES3]
	Recognition and integration of the Pharmaceutical Department in the Health Secretariat	“*We used to go to the administrative meetings but when there was a more technical meeting sometimes the pharmacy was forgotten. So, we make them remember us and started to be more informed of the routines of the unit, of how things worked the interaction with the other professionals improved very much and this was also a benefit to us. To the service.*” [ESE4]
	Sustainability of the management actions	“*We elaborated a part of the Municipal Health Plan focused on the pharmaceutical policy. We got all sectors to participate in the construction of the Municipal Health Plan and to be engaged in the execution of what was in the Plan. This brought visibility and autonomy to the area.*” [ENE7]
	Incorporation of the use of management tools in the practices of institutions	“*Some of the tools that we came across in the course I already knew of, but I was only able to apply them and use them in real life in this course.*” [GF05] “*I used the* [SSP] *techniques that I learned to computerize the entire execution of specialized attention services. Our planning was all computerized.*” [ECO5]

## Data Availability

The data presented in this study are available on request from the corresponding author.

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
