# Peer review of "Attending a Blended In-Service Management Training in a Public Health System: Constraints and Opportunities for Pharmacists and Health Services"

_pharmacy, 2021, doi:10.3390/pharmacy9010012_

Round 1
Reviewer 1 Report
-Although you did a great job overall describing the context in the introduction, I think potentially, it could have been shortened a bit or some of the points could have been combined. This would keep the readers engaged with the topic.
-Lines 95 to 98 in the methods potentially belong more in the discussion. Reading initially, I was guessing this was to explain potential barriers or challenges but it's not really methods.
-For the methods, I would have liked to know more about the creation of the materials used and in particular how the order and the spacing of content was decided. What was deemed online and what was selected for print only? I feel that this could impact the course success and retention of materials. How many of each areas of course themes were included?
-For the methods, including the number of pharmacists who completed and enrolled in the program should be moved to the results.
-For the theoretical models' section, I believe this could be stated more concisely.
-Table 1 was very helpful to describe the survey/assessment piece.
-I would have liked to know more about the questionnaire design. Was it pilot tested? How was it administered? What were the instructions provided? Was interrater reliability established?
-Minutes of focus groups belong in the results section.
-I believe that perhaps the assessment methods could be presented as a series of papers. There is a lot of information here and I think that presenting all in one paper may have been part of the challenge in making this clear and concise.
Author Response
Letter to the Reviewers
Dear Editors of Pharmacy
It was with a great satisfaction that we received the comments in our manuscript entitled “Attending a blended in-service management training in a public health system: constraints and opportunities for pharmacists and health services”. We appreciated the opportunity to revise it. The comments and suggestions are important and helped us to improve the manuscript.
We would like to highlight that a professional language specialist reviewed the complete text with the aim to clarify the text and enhance the English version quality. The changes related to the language correction are not highlighted in text to avoid visual pollution.
Below are the responses to the reviewers' comments and contributions attributed to the manuscript highlighted in the main text:
Reviewer 1
- Although you did a great job overall describing the context in the introduction, I think potentially, it could have been shortened a bit or some of the points could have been combined. This would keep the readers engaged with the topic.
The Introduction text was shortened cutting some phrases (lines 45-46, 52-53, 57-58, 84-85).
- Lines 95 to 98 in the methods potentially belong more in the discussion. Reading initially, I was guessing this was to explain potential barriers or challenges but it's not really methods.
We understand that it is not a discussion, but the context in which the course was developed. So, we decided to keep the lines in the text.
- For the methods, I would have liked to know more about the creation of the materials used and in particular how the order and the spacing of content was decided. What was deemed online and what was selected for print only? I feel that this could impact the course success and retention of materials. How many of each areas of course themes were included?
A table (Table 1, line 117) has been added to better describe the course and its contents.
- For the methods, including the number of pharmacists who completed and enrolled in the program should be moved to the results.
The number of students who enrolled and attended the course, distinguishing between classes A and B, are important information about the studied course. They are not results of this study, but the course’s context. So, we decided to keep these information in item 2.1. Environment and context of the course
- For the theoretical models' section, I believe this could be stated more concisely.
Text was shortened (lines 131-144).
- I would have liked to know more about the questionnaire design. Was it pilot tested? How was it administered? What were the instructions provided? Was interrater reliability established?
An explanation was inserted in the lines 113-184.
- Minutes of focus groups belong in the results section.
It was moved to the line 286.
- I believe that perhaps the assessment methods could be presented as a series of papers. There is a lot of information here and I think that presenting all in one paper may have been part of the challenge in making this clear and concise.
The intention is to provide a broader understanding of the students' perception of the course, so we sought to compile the results of several evaluation strategies developed over the course.
Reviewer 2 Report
Thank you for the opportunity to review this work which is an interesting topic contextualised to Brazilian health system, in particular pertaining to pharmacists working in the public health system.
The writing is sometimes a little convoluted with long sentences. A thorough proof-read is recommended to help with readability.
Overall, I appreciate that the work although focussed on a specific health system, could be of interest to a wider audience in the area of CPD or CE.
Please see below some more specific comments for consideration.
Abstract:
The abstract is not well ordered and does not have the expected detail. The information for the background is fine. However, in the methods section, it should be clear what was undertaken with number of participants where appropriate and some indication of how data was analysed. In the results again numbers/percentages should be reported for quantitative data. There are no obvious conclusions.
Introduction:
Line 39 does not really make sense. I don't really understand what this means.
Study design:
The description of the theoretical basis of the study appears well considered. However, it is not clear how the theories and models have been used to inform the data collection and analysis. This needs to be more clearly articulated.
I have not seen focus groups termed 'focal' groups before. I would suggest sticking with well-accepted terminology.
line 203-204: this detail should be included in the results
line 209: total no. of mins of audio is results
Results
Lines 240-245: These are just lists of variables and not formal sentences.
Table 3: the issues addressed are phrased as questions that would be answered dichotomously rather than with the agree/disagree options. Please consider rephrasing appropriately.
Some of the methods are repeated in the results unnecessarily. Can the authors explain why focus groups and interviews were conducted 3 yrs later.
Discussion:
There is a lot of reporting of the results in the discussion. It is acknowledged that there is some contextualisation to the wider literature. However, the discourse could be more succinct to highlight the key learning of the study rather than revisiting all the findings.
There is no final summary or conclusions which is expected at the end of a study to sum up the significance and/or future recommendations.
Author Response
Letter to the Reviewers
Dear Editors of Pharmacy
It was with a great satisfaction that we received the comments in our manuscript entitled “Attending a blended in-service management training in a public health system: constraints and opportunities for pharmacists and health services”. We appreciated the opportunity to revise it. The comments and suggestions are important and helped us to improve the manuscript.
We would like to highlight that a professional language specialist reviewed the complete text with the aim to clarify the text and enhance the English version quality. The changes related to the language correction are not highlighted in text to avoid visual pollution.
Below are the responses to the reviewers' comments and contributions attributed to the manuscript highlighted in the main text:
Reviewer 2:
Abstract: The abstract is not well ordered and does not have the expected detail. The information for the background is fine. However, in the methods section, it should be clear what was undertaken with number of participants where appropriate and some indication of how data was analysed. In the results again numbers/percentages should be reported for quantitative data. There are no obvious conclusions.
The text was modified to answer the suggestions. However, the number of words defined (200 words) limits the inclusion of all the information asked.
Introduction: Line 39 does not really make sense. I don't really understand what this means.
The phrase was modified (line 36).
Study design: The description of the theoretical basis of the study appears well considered. However, it is not clear how the theories and models have been used to inform the data collection and analysis. This needs to be more clearly articulated.
The study groups the data were defined into three categories: a) students’ sociodemographic, educational, and professional characteristics – considering the Theoretical Domains Framework, particularly the capacity and motivation domains; b) evaluation of the infrastructure, contents, and teaching methods – considering the Theoretical Domains Framework domain of opportunity, and the Kirkpatrick’s level of satisfaction with the learning intervention; c) perceived impacts on the students’ behaviors and the management practices in health services – considering the domain of opportunity (related to the workplace conditions) and the 2º, 3º and 4º levels of evaluation from Kirkpatrick’s model. All of the analyses also considered the Brazilian Policy for health professionals’ permanent education in the SUS, so the definition of the categories was in line with the premises that the impact of a continuing education intervention is a result of the confluence between students, workplace and education program.
We think this explanation is already described in the lines 125-154.
I have not seen focus groups termed 'focal' groups before. I would suggest sticking with well-accepted terminology.
It was a mistake. The text was modified (lines 191-217)
line 203-204: this detail should be included in the results
We believe this information is part of the description of the sample.
line 209: total no. of mins of audio is results
It was moved to the line 286.
Results: Lines 240-245: These are just lists of variables and not formal sentences.
The lines were moved to the legend of the Table 3 (lines 263-266).
Table 3: the issues addressed are phrased as questions that would be answered dichotomously rather than with the agree/disagree options. Please consider rephrasing appropriately.
We chose methodologically to separate according to the degree of agreement. To reformulate, would mean to reapply the form, which is no longer viable, at this point.
Some of the methods are repeated in the results unnecessarily. Can the authors explain why focus groups and interviews were conducted 3 yrs later.
The text has been adjusted. The new analysis, after three years, was carried out to assess the long-term impact. It was included in the lines 195-197.
Discussion: There is a lot of reporting of the results in the discussion. It is acknowledged that there is some contextualisation to the wider literature. However, the discourse could be more succinct to highlight the key learning of the study rather than revisiting all the findings.
The 5 first paragraphs were rewrote and shortened to answer the suggestion.
There is no final summary or conclusions which is expected at the end of a study to sum up the significance and/or future recommendations.
Final remarks were added to the text, lines 416-427
Reviewer 3 Report
This manuscript describes the impact of management training course in Brazilian pharmacists.
The topic of the study is met of journal’s scope and I recommend that the manuscript is necessary to be revised:
1. Word selection: I was sometimes confused to understand the context of the manuscript because of word selection.
1-1 “Pharmacists” and “Students” seemed to use in the same sense. I confused they have same meaning or not.
1-2 Please explain what are “class A” and “class B”. I didn’t understand what they are meaning.
1-3 Mixed method/multi-method. Do they have the same meaning? I understand mixed method is a kind of statistical method but I am not sure for multi-method.
1-4 They interviewed by Portuguese. How did you validate of translation from Portuguese to English and vice versa. Please comment.
2. Study design: In line 1999, authors indicated new students were randomly selected until desired sample size was reached. However this process may lead some bias if many students refuse to receive interviewing. Please comment.
3. Tables
Table 1-3, authors provided only percentage of each category. I recommend to adding the number of responders. I am thinking authors used chi-square test (should be addressed in the manuscript) to calculate the p-values in table 3 and the number of some cells would be quite low column each category. If so, Fisher’s exact test instead of chi-square test would be desirable.
4. Mixed method: The authors used both quantitative and qualitative data, however integration of two different data sources seems to be insufficient as a mixed method. I feel the authors discussed using quantitative and qualitative data independently. I recommend discussing with integration of quantitative and qualitative data more, if the author insist a mixed method.
Author Response
Letter to the Reviewers
Dear Editors of Pharmacy
It was with a great satisfaction that we received the comments in our manuscript entitled “Attending a blended in-service management training in a public health system: constraints and opportunities for pharmacists and health services”. We appreciated the opportunity to revise it. The comments and suggestions are important and helped us to improve the manuscript.
We would like to highlight that a professional language specialist reviewed the complete text with the aim to clarify the text and enhance the English version quality. The changes related to the language correction are not highlighted in text to avoid visual pollution.
Below are the responses to the reviewers' comments and contributions attributed to the manuscript highlighted in the main text:
Reviewer 3:
Word selection: I was sometimes confused to understand the context of the manuscript because of word selection.
The entire text has undergone a professional review of the English language.
“Pharmacists” and “Students” seemed to use in the same sense. I confused they have same meaning or not.
The text was adjusted, it was decided to use “students”.
Please explain what are “class A” and “class B”. I didn’t understand what they are meaning.
The text was adjusted in the lines 98-99. The PSAMM Coure was offered in two groups, called A and B (in different times). This information is important just to emphasize that the sample for some parts of this study was limited to group B.
Mixed method/multi-method. Do they have the same meaning? I understand mixed method is a kind of statistical method but I am not sure for multi-method.
We used mixed-methods, meaning qualitative and quantitative data were collected and analysed. They were integrated in the interpretation and discussion phase (lines 232-234).
They interviewed by Portuguese. How did you validate of translation from Portuguese to English and vice versa. Please comment.
The data collection, analyzes and interpretations were carried out in the native language (Portuguese) of researchers and interiewees. The translation was performed only for the presentation of the manuscript, by a professional person.
Study design: In line 1999, authors indicated new students were randomly selected until desired sample size was reached. However this process may lead some bias if many students refuse to receive interviewing. Please comment.
As the sample was random, the call for new students previously selected and who could not participate, does not impact as bias.
Tables: Table 1-3, authors provided only percentage of each category. I recommend to adding the number of responders. I am thinking authors used chi-square test (should be addressed in the manuscript) to calculate the p-values in table 3 and the number of some cells would be quite low column each category. If so, Fisher’s exact test instead of chi-square test would be desirable.
The tables have been adjusted. Fisher's test was applied.
Mixed method: The authors used both quantitative and qualitative data, however integration of two different data sources seems to be insufficient as a mixed method. I feel the authors discussed using quantitative and qualitative data independently. I recommend discussing with integration of quantitative and qualitative data more, if the author insist a mixed method.
We understand we have brought qualitative and quantitative data together define the categories of analyses, to interpret the results and to identify the conclusions. The integration of the data was important to identify possible convergent and divergent results and in this study, was particularly useful to complement the finds. It is possible to see the integration in the discussion: data presented in the tables 4,5,6 and 7 were interpreted and discussed together and allowed the identification of the conclusions.
Round 2
Reviewer 2 Report
The previous comments have been largely addressed. The language is still somewhat verbose and convoluted but the reader can follow the discourse.
There is just one discrepancy:
In the abstract, it states 60 focus groups were undertaken (n=60), when in fact 10 focus groups were undertaken with 6 participants, so it should read (n=10).
Author Response
Dear Reviewer
Thank you for your thoughtful revision of the text. The error in the number of focus groups in the abstract has been corrected.